# Predicting Cyanobacterial Harmful Algal Blooms (CyanoHABs) in a Regulated River Using a Revised EFDC Model

**Jung Min Ahn** , **Jungwook Kim \*** , **Lan Joo Park, Jihye Jeon, Jaehun Jong, Joong-Hyuk Min and Taegu Kang**

Water Quality Assessment Research Division, Water Environment Research Department, National Institute of Environmental Research, Incheon 22689, Korea; ahnjm80@gmail.com (J.M.A.); mintaka35@korea.kr (L.J.P.); jhjeon16@korea.kr (J.J.); jongjaehun@korea.kr (J.J.); joonghyuk@korea.kr (J.-H.M.); taegu98@korea.kr (T.K.)
\* Correspondence: rlawjddnr1023@gmail.com

**Abstract:** Cyanobacterial Harmful Algal Blooms (CyanoHABs) produce toxins and odors in public water bodies and drinking water. Current process-based models predict algal blooms by modeling chlorophyll-a concentrations. However, chlorophyll-a concentrations represent all algae and hence, a method for predicting the proportion of harmful cyanobacteria is required. We proposed a technique to predict harmful cyanobacteria concentrations based on the source codes of the Environmental Fluid Dynamics Code from the National Institute of Environmental Research. A graphical user interface was developed to generate information about general water quality and algae which was subsequently used in the model to predict harmful cyanobacteria concentrations. Predictive modeling was performed for the Hapcheon-Changnyeong Weir–Changnyeong-Haman Weir section of the Nakdong River, South Korea, from May to October 2019, the season in which CyanoHABs predominantly occur. To evaluate the success rate of the proposed model, a detailed five-step classification of harmful cyanobacteria levels was proposed. The modeling results demonstrated high prediction accuracy (62%) for harmful cyanobacteria. For the management of CyanoHABs, rather than chlorophyll-a, harmful cyanobacteria should be used as the index, to allow for a direct inference of their cell densities (cells/mL). The proposed method may help improve the existing Harmful Algae Alert System in South Korea.

**Keywords:** water quality modeling; harmful cyanobacteria; CyanoHABs; EFDC-NIER

## 1. Introduction

An algal bloom is a phenomenon in which there is an abnormal proliferation of photosynthetic algae in water bodies which turns the water in a river or lake green. From the perspective of traditional taxonomy, algal blooms can be caused by green algae, diatoms, or cyanobacteria. Algal blooms can occur globally. In South Korea, this phenomenon is mainly caused by cyanobacteria. Cyanobacterial Harmful Algal Blooms (CyanoHABs) that occur in freshwater lakes, rivers, and estuaries are caused by cyanobacteria, which are also known as blue-green algae, and occur most often in the summer. Species belonging to the genus *Microcystis* spp. are representative of CyanoHABs in South Korea's freshwaters [1]. CyanoHABs cause many social, economic, and environmental problems each summer in South Korea. Aquatic species, such as fish, are killed by toxins produced by *Microcystis* spp. [2–4], eventually leading to the degradation of the aquatic ecosystem. It also adversely affects the water management of the drinking water protection zone. The United States Environmental Protection Agency stated, "CyanoHABs and their toxins can harm people, animals, aquatic ecosystems, the economy, drinking water supplies, property values, and recreational activities, including swimming and commercial and recreational fishing in many countries". This means that the prediction and management of CyanoHABs is imperative not only in South Korea, but also globally. Therefore, it is necessary to predict the occurrence of harmful cyanobacteria, *Microcystis* spp., *Anabaena*(=*Dolichospermum)* spp.,

and *Aphanizomenon* spp., for improved water quality management in countries where CyanoHABs often occur.

Chung and Lee [5] calculated the "occupancy ratio by algal group" (i.e., by each algal type) based on measured values of algal cell numbers and simulated algal blooms by converting the measured chlorophyll-a concentrations to concentrations of each algal group. Yajima and Choi [6] simulated algal patterns by calculating the contribution of each algal group to the total chlorophyll-a using the chlorophyll-a content (pg/cell) per unit cell. The authors used the cell numbers provided by Reynolds [7] for the dominant species in each major algal group found in Lake Urayama in Japan. Chung et al. [8] determined the chlorophyll-a content per unit cell of *Microcystis* spp. ($1.07 \times 10^{-6}$ μg Chl-a/cell) and the cell number in Daecheong Lake based on the chlorophyll-a concentration, and then compared this cell number with the measured cell number. Then, they used the chlorophyll-a concentration as a proxy and predicted algal blooms in the Nakdong River using the Environmental Fluid Dynamics Code (EFDC), which is a three-dimensional (3D) water quality model from the National Institute of Environmental Research (NIER), thereby improving the prediction accuracy from existing models [9]. To investigate the main cause of algal blooms in Xiangxi Bay, the correlation between seawater and chlorophyll-a concentration was analyzed, and algal blooms were simulated using the EFDC [10]. Since cyanobacteria also have chlorophyll, chlorophyll-a is widely used in models simulating cyanobacteria.

Chlorophyll-a is generated by diatoms, green algae, and cyanobacteria. Therefore, using chlorophyll-a as a water quality management index in countries affected by CyanoHABs caused by harmful cyanobacteria is an issue [11]. Despite the occurrence of CyanoHABs, the Harmful Algae Alert System was not activated because the chlorophyll-a concentration was lower than the standard value when chlorophyll-a was previously the criterion for the Harmful Algae Alert System. Therefore, chlorophyll-a, which is not correlated with harmful cyanobacteria, was omitted from the standard, and the Harmful Algae Alert System operated with only the harmful cyanobacteria cell number in South Korea. Lake Erie, one of the Great Lakes in North America, has a similar CyanoHABs problem as that in South Korea. Millie et al. [12] generated robust ecological niche models for *Microcystis* spp. using Artificial Neural Networks and provided a predictive framework for quantitative visualization of nonlinear CyanoHAB–environment interactions. In Lake Erie, *Microcystis* spp. generally dominates from mid-July through early November, which corresponds to summer [13]. Studies to estimate the concentration and distribution of *Microcystis* spp. were carried out to predict the occurrence of CyanoHABs in Lake Erie [14,15]. *Microcystis* spp. abundance and presence are usually underestimated in traditional phytoplankton quantifications because they have large colony size and relatively low numerical abundance [16]. Therefore, it is important to accurately quantify *Microcystis* spp. for the predictability of CyanoHABs.

A significant correlation between harmful cyanobacteria and concentration of chlorophyll-a in South Korea's freshwater environment has not been found [17]. Therefore, it is reasonable to use the number of harmful cyanobacteria cells as an indicator for predicting the occurrence of CyanoHABs. We propose a technique that can accurately predict CyanoHABs using the EFDC-NIER source code. Since species with different occurrence and behavior characteristics are mixed within the same algae group, there may be limitations in reproducing the rapid occurrence of specific species and complex species transitions. Therefore, a process was developed to independently simulate each phylum or genus to complement the process of simulating algae in three groups (e.g., cyanobacteria, green algae, and diatom). For example, only *Microcystis* spp. can be simulated separately in cyanobacteria. Thus, it is possible to independently predict the occurrence of harmful cyanobacteria. The aims of this study are as follows (Figure 1):

(1)　To develop a Graphical User Interface (GUI) that automatically generates input data for EFDC-NIER modeling to improve the ease of prediction of CyanoHABs.

(2) To predict CyanoHABs by applying the GUI and using the EFDC-NIER to the section between Hapcheon-Changnyeong Weir and Changnyeong-Haman Weir, where severe occurrences of HABs are observed.

(3) To verify the accuracy of the prediction of CyanoHABs and suggest improvements to the Harmful Algae Alert System.

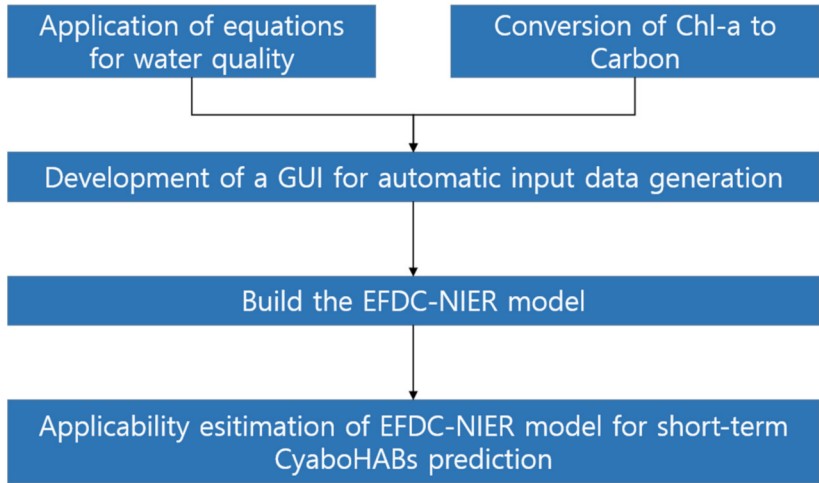

**Figure 1.** Workflow process of the study.

## 2. Materials and Methods

### 2.1. EFDC-NIER Model

The EFDC was developed by the Virginia Institute of Marine Science in the U.S., and the U.S. Environmental Protection Agency (EPA) released the Generalized Vertical Grid version. Then, DSI released EFDC_DS (version 20100328) and has continuously updated the source code. The water quality parameters of the EFDC include information on phytoplankton, carbon, nitrogen, phosphorus, and silicon cycles, and dissolved oxygen and chemical oxygen demand (COD). Phytoplankton are divided into three algal group (diatoms, green algae, and cyanobacteria). This division makes it convenient to account for seasonal transitions. Since algal groups with different behavioral characteristics are mixed in the same algal group, there may be limitations in reproducing the rapid occurrence of certain algae and complex species transitions. Therefore, to model multiple algal species, the NIER improved the source code of the model. As shown in the schematic of the reactions among multiple algal species (CHx1–CHxn) in Figure 2, the algae-related state parameters were expanded based on the EFDC_DS (version 20100328) (red indicates where the mechanism of algae generation and death affects (Figure 2)).

The extended reaction Equation (1) for multiple algal species is identical to the reaction equation written for three algal species in the earlier EFDC model:

$$\frac{\partial B_x}{\partial t} = (P_x - BM_x - PR_x)B_x + \frac{\partial}{\partial z}(WS_x \cdot B_x) + \frac{WB_x}{V} \tag{1}$$

where t is time (day), V is the grid volume ($m^3$), $B_x$ is the biomass ($gC/m^3$), $P_x$ is the growth rate ($day^{-1}$), $BM_x$ is the metabolic rate ($day^{-1}$), $PR_x$ is predation rate ($day^{-1}$), $WS_x$ is sedimentation rate ($day^{-1}$), and $WB_x$ is the external inflow of the algal species x (gC/day).

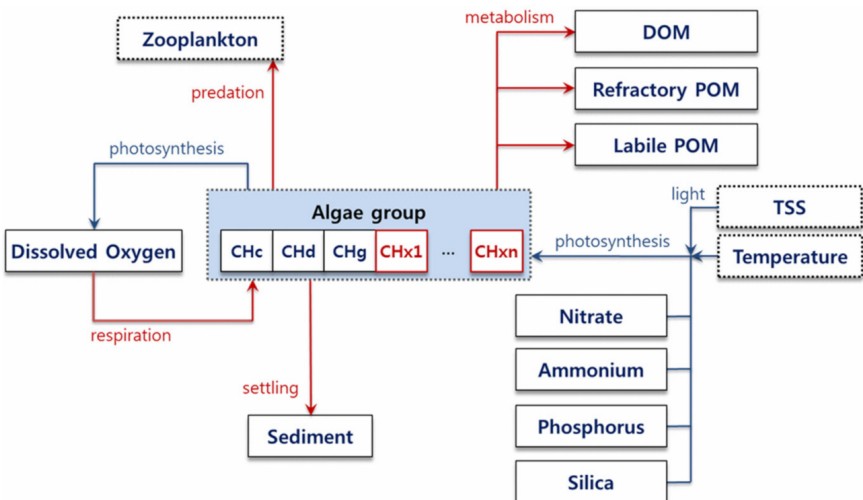

**Figure 2.** Schematic of the reactions among multiple algal species (CHc: cyanobacteria, CHd: diatoms, CHg: green algae, CHx1–CHxn: multiple algal species, DOM: dissolved organic material, POM: particulate organic material [18].

*2.2. Application of Input Values to the EFDC-NIER Model*

2.2.1. Application of Equations for General Water Quality

The EFDC-NIER model requires the following water quality observation data: Total Nitrogen (T−N) (mg/L), Nitrate Nitrogen (NO$_3$−N) (mg/L), Ammonia Nitrogen (NH$_3$−N) (mg/L), Total Phosphorus (T−P) (mg/L), Water temperature (°C), Dissolved Total Nitrogen (DTN) (mg/L), Dissolved Total Phosphorus (DTP) (mg/L), Phosphate (PO$_4$-P) (mg/L), Chlorophyll-a (mg/m$^3$), Biochemical Oxygen Demand (BOD) (mg/L), Chemical Oxygen Demand (COD) (mg/L), Dissolved Oxygen (DO) (mg/L), and Total Organic Carbon (TOC) (mg/L). From these data, the following variables were estimated using the equations presented in NIER [19] (Table 1): refractory particulate organic carbon (ROC), labile particulate organic carbon (LPOC), dissolved organic carbon (DOC), refractory particulate organic phosphorus (RPOP), labile particulate organic phosphorus (LPOP), dissolved organic phosphorus (DOP), refractory particulate organic nitrogen (RPON), labile particulate organic nitrogen (LPON), and dissolved organic nitrogen (DON).

**Table 1.** Conversion equations for water quality observation data to be used as input data in the EFDC-NIER model.

| Series | Water Quality Variable | Modeling Variable | Input Data Equations |
|---|---|---|---|
| Carbon | TOC | RPOC | $=(DOC - OC) \times 0.5$ |
| | | LPOC | $=(DOC - OC) \times 0.5$ |
| | | DOC | $=(BOD - AOD_5 - NOD_5)/(1 - e^{-5 \times Kdbot\,*}) \times (12/32)$ |
| Nitrogen | TN NH$_4$-N NO$_3$-N DTN | RPON | $=(TN - Algae\ Nitrogen - DTN) \times 0.5$ |
| | | LPON | $=(TN - Algae\ Nitrogen - DTN) \times 0.5$ |
| | | DON | $=DTN - NH_4 - NO_3$ |
| | | NH$_4$ | $=NH_4$ |
| | | NO$_3$ | $=NO_3$ |
| Phosphorous | TP PO$_4$-P DTP | RPOP | $=(TP - Algae\ Phosphorus - DTP) \times 0.5$ |
| | | LPOP | $=(TP - Algae\ Phosphorus - DTP) \times 0.5$ |
| | | DOP | $=DTP - PO_4$ |
| | | PO$_4$ | $=PO_4$ |

* Kdbot: Organic substance decomposition rate in the BOD bottle (day$^{-1}$).

### 2.2.2. Conversion of Chlorophyll-a Concentration to Carbon Content

The total carbon content of each algal group (diatoms, green algae, and cyanobacteria) in each tributary that flows into a main stream is required to provide boundary conditions. The best strategy would be to convert the cell counts of algal species—observed over a long period at the inflow of each tributary—into a carbon amount. However, for the Nakdong River, cell count data for the tributaries were not available. Therefore, the chlorophyll-a contents at the tributary endpoints were converted to a carbon content for each harmful algal bloom prediction group using cell count data collected at seven-day intervals from a point located 500 m upstream from the Changnyeong-Haman Weir (in the main stream) in 2018. The sampling method is shown in Appendix A Figure A1. Classification of algal species according to characteristics such as habitat environment, environmental tolerance, and sensitivity is called 'Codon' [20] (Figure 3).

**Figure 3.** Procedure for calculating monthly carbon occupancy proportions of each algal group [18].

The following steps were used for this conversion (as outlined in Figure 3):

1. The 684 species of algae observed in the Nakdong River were classified into five algal groups and their carbon content per cell was determined (Table 2).
2. The carbon content per liter was determined for each species. For example, for *Asterionella* spp., the number of observed cells was 240 cells/mL and the carbon content per cell was 125 pg C/cells. Therefore, the carbon content per liter was calculated as follows: 240 cells/mL × 125 pg C/cells = 240 cells/mL × 125 × 10$^{-9}$ mg C/cells = 0.00003 mg C/mL = 0.03 mg C/L.
3. Based on the number of cells observed every week, the carbon occupancy rate was calculated for each of the five groups.
4. The carbon occupancy proportions for each of the five groups per month were calculated (Table 3).
5. Under the assumption that the carbon occupancy proportions of each harmful algal bloom prediction group observed in the main stream were identical to those in the tributaries, the chlorophyll-a observed in the tributaries was converted to a carbon content for each harmful algal bloom prediction group using the monthly carbon occupancy proportions and the carbon:chlorophyll-a ratio ($\beta$; see below).

In the Changnyeong-Haman Weir, the average ratio of carbon content to observed chlorophyll-a concentration for each prediction group from 2013 to 2018 was 0.12, and the average ratio for eight locations monitored in the Nakdong River (Sangju, Nakdan, Gumi, Chilgok, Gangjeong-Goryeong, Dalseong, Hapcheon-Changnyeong, and Changnyeong-Haman weirs) was also 0.12. Therefore, 0.12 was used as $\beta$ in this study. Appendix B Figure A4 compares the observed chlorophyll-a with that calculated using $\beta$ = 0.12.

**Table 2.** Classification of algal species observed in the Nakdong River into groups and their respective carbon content per cell.

| Group | | Species (pgC/cell) |
|---|---|---|
| Cyano bacteria | Group M | *Microcystis* spp. (10.95) |
| | Group H1 | *Anabaena(=Dolichospermum)* spp. (164.1), *Aphanizomenon* spp. (9.5) |
| Diatoms | | *Nitzschia* spp. (56.2), *Skeletonema* spp. (127.8), *Stephanodiscus* spp. (520.5), *Synedra* spp. (516.1), *Aulacoseira* spp. (201.3), *Fragilaria* spp. (68.9), *Melosira* spp. (705.1), *Closteriopsis* spp. (130.4), *Closterium* spp. (143.5), *Staurastrum* spp. (13,651.8), *Asterionella* spp. (125.2), *Cyclotella* spp. (301.5) |
| Green algae | | *Chroomonas* spp. (407.5), *Cryptomonas* spp. (407.5), *Chlamydomonas* spp. (446.5)*Carteria* spp. (27.1), *Eudorina* spp. (161.6), *Pandorina* spp. (204.4), *Actinastrum* spp. (9.3), *Coelastrum* spp. (123.4), *Crucigenia* spp. (12.9), *Golenkinia* spp. (54.7), *Pediastrum* spp. (8.5), *Scenedesmus* spp. (10.6), *Tetraedron* spp. (90.4), *Tetrastrum* spp. (6.4) |
| Other algae | | *Ceratium* spp. (361.8), *Gymnodinium* spp. (1,303.15), *Peridinium* spp. (2,244.5), *Merismopedia* spp. (0.4) |

Equation (2) was used to convert the chlorophyll-a concentration observed in the main stream into a total carbon content to be used as a boundary condition for the inflow from the tributary. After the chlorophyll-a concentration was converted to a carbon content for each harmful algal bloom prediction group using $\beta = 0.12$ mg C/µg chl-a, the algal carbon content at each boundary was added to the monthly proportion of each algal group (based on the carbon content) of the Changnyeong-Haman Weir in Table 3. For example, if the observed chlorophyll-a concentration in July in the Changnyeong-Haman Weir was 40 mg/m$^3$, the carbon amount of *Microcystis* spp. was $0.12 \times 0.281 \times 40 = 1.3488$ mg C/L.

$$X_{all} \text{ carbon} = \sum_{i=1}^{9} (\beta \times X_i \text{carbon occupancy proportion} \times \text{chlorophyll—aconcentration}) \tag{2}$$

where $X_i$ is the $i_{th}$ algal group and $\beta = 0.12$.

**Table 3.** Monthly carbon occupancy proportions (%) for each algal group in the Changnyeong-Haman Weir.

| Group | Jan | Feb | Mar | Apr | May | Jun | Jul | Aug | Sep | Oct | Nov | Dec |
|---|---|---|---|---|---|---|---|---|---|---|---|---|
| Group M * | 0.0 | 0.0 | 0.0 | 0.0 | 0.1 | 4.2 | 28.1 | 41.5 | 7.8 | 0.2 | 0.0 | 0.0 |
| Group H1 * | 0.0 | 0.0 | 0.0 | 0.0 | 3.8 | 1.1 | 3.0 | 1.2 | 3.1 | 1.2 | 0.7 | 0.4 |
| Diatoms | 72.7 | 85.0 | 89.6 | 50.3 | 41.9 | 76.8 | 34.6 | 21.7 | 36.7 | 24.4 | 63.7 | 62.9 |
| Green algae | 25.4 | 14.7 | 10.3 | 49.5 | 54.1 | 17.8 | 29.7 | 30.2 | 49.5 | 73.8 | 35.6 | 36.7 |
| Other algae | 1.9 | 0.3 | 0.1 | 0.2 | 0.0 | 0.1 | 4.6 | 5.4 | 3.0 | 0.4 | 0.0 | 0.0 |

* Cyanobacteria.

### 2.2.3. Development of a GUI for Automatic Input Data Generation

In order to consistently and rapidly convert the water quality data collected by the Water Environment Information System [21] into data appropriate for the EFDC-NIER model using the method described in Section 2.2.2, a GUI was developed using Microsoft Excel's Visual Basic tool (Figure 4). The Water Environment Information System is a freely downloadable database that provides all water quality observational data in South Korea. The GUI was composed of the following modules:

1. A storage module for storing the carbon occupancy proportions of each harmful algal bloom prediction group in each period.
2. An input module for receiving the water quality data observed at the tributary endpoints, used for boundary conditions.
3. A conversion module for converting the chlorophyll-a concentrations to carbon contents for each harmful algal bloom prediction group at the observation time (to enable modeling by matching these amounts with the carbon occupancy proportions of each harmful algal bloom prediction group according to the modeling period).

4. A modeling module for 3D numerical modeling of each harmful algal bloom prediction group for all study areas based on the carbon content of each harmful algal bloom prediction group.

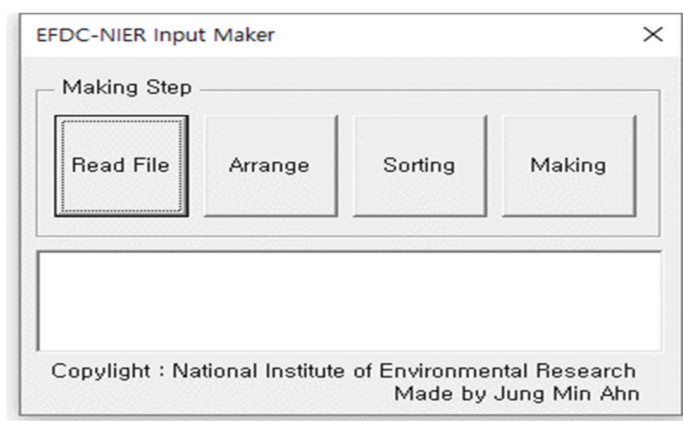

**Figure 4.** Automatic input data generation module for EFDC-NIER modeling.

*2.3. Building and Modeling of the EFDC-NIER Model*

As shown in Figure 5, there are eight multifunctional weirs in the Nakdong River, which is subdivided into 22 mid-sized basin units. In the Nakdong River, problems related to CyanoHABs are common. In particular, the weirs in the downstream portion of the river have poor water quality and more frequent CyanoHABs compared with those in the upstream portion. The Hapcheon-Changnyeong Weir–Changnyeong-Haman Weir section (72 km) was selected for this study. It is an appropriate study area because CyanoHABs continuously occur in summer; moreover, this section serves as a water source. As shown in Figure 5, the EFDC-NIER model was applied to the Hapcheon-Changnyeong Weir and the Samrangjin water level observatory section. The factors influencing the water balance, such as tributaries flowing into the main stream, effluent from sewage treatment plants, and water intake stations, were accounted for in the model as boundary conditions. The number of horizontal and vertical grid elements was 11,290 and 5, respectively. In addition, 15 transverse grids were used to reflect the water body flow with respect to the operation of hydraulic structures. The "Mask" option was set for the grids where multifunctional weirs were located, and the water flowing upstream was discharged to the downstream section using the hydraulic structure module (QCTL) (Figure 6). The following data were used in the model: hourly meteorological observation data from the Meteorological Data Open Portal of the Korea Meteorological Administration [22]; daily weir operation data provided by K-water [23]; daily dam discharge data provided by the Water Resource Management Information System [24]; flow observation data provided by the Ministry of Environment; and water quality monitoring network data from the Ministry of Environment [21].

In this study, modeling of general water quality variables and algae-related water quality was performed for the study area using the model input data generation GUI developed in Section 2.2. The Hapcheon-Changnyeong Weir was set as the upstream boundary of the model, whereas the Samrangjin water level observatory section was chosen as the downstream boundary. Within the model domain, the Changnyeong-Haman Weir was included as a hydraulic structure. The reproducibility of the model was evaluated for the period from 1 May to 30 October 2019. Predicted values were compared with observed values collected at a point located 500 m upstream from the Changnyeong-Haman Weir. The hydraulic structures in the Changnyeong-Haman Weir were as follows (from the left to the right bank): a fishway, a fixed weir, a movable weir, another fixed weir, a small hydropower station, and finally another fishway. The discharge priorities for the operation of the hydraulic structures are as follows: fishway $\geq$ small hydropower station > movable weir > fixed weir. During the modeling period, the movable weirs

were operated in turning type, and most flows were discharged through the fishway and small hydropower station. Thus, the flow of the water body was concentrated near the right bank.

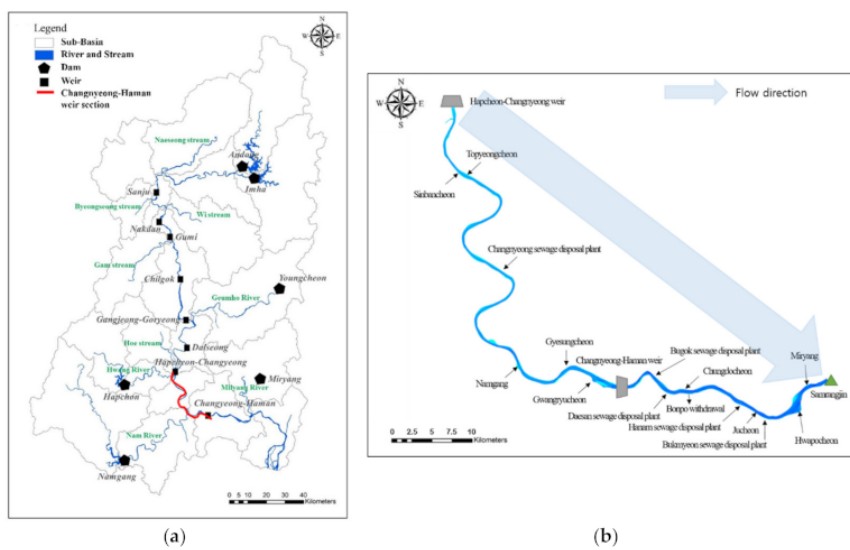

**Figure 5.** Study area: (**a**) Nakdong River Basin; (**b**) Hapcheon-Changnyeong weir-Changnyeong-Haman weir section.

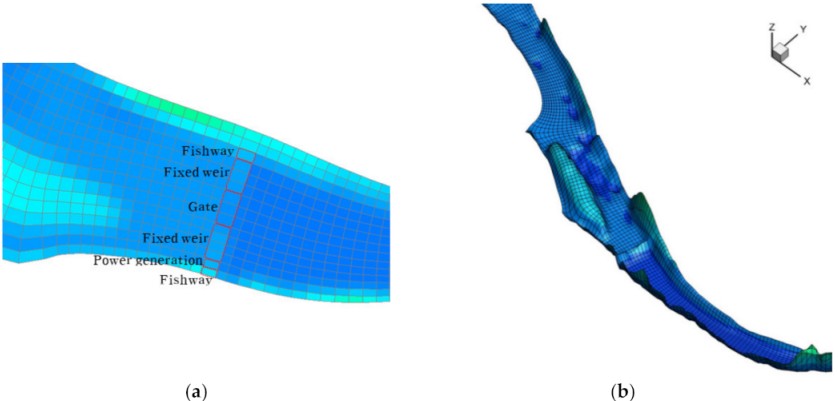

**Figure 6.** Changnyeong-Haman Weir: (**a**) Application of multifunctional weir hydraulic structure module; (**b**) Terrains around the Changnyeong-Haman Weir.

The model parameters were calibrated using the mean absolute error (MAE) and the root mean square error (RMSE). The MAE is the average of the absolute errors between the observed and simulated values and can be used to compare the residuals between models. The RMSE is a mean error between the observed and simulated values and serves as an indicator of model precision. The values in Table 4 were used as the major parameters for harmful algal bloom and water quality analysis. The RMSE and MAE in the model simulation period are summarized in Table 5. The equations for MAE and RMSE are as follows:

$$MAE = \frac{\sum_{i=1}^{N} |O_i - P_i|}{N} \tag{3}$$

and

$$RMSE = \sqrt{\frac{1}{N} \sum_{i=1}^{N} (O_i - P_i)^2} \tag{4}$$

where $P_i$ is the simulated value at time i, $O_i$ is the observed value at time i, and N is the number of observed values in the whole period.

**Table 4.** Major parameters applied to the EFDC-NIER model [18].

| EFDC Parameter | | Unit | Definition | Nakdong River |
|---|---|---|---|---|
| PM$_x$ | Group M | d | Max. growth rate | 2.0 |
| | Group H1 | | | 2.0 |
| | Diatom | | | 2.0 |
| | Green | | | 2.0 |
| | Other | | | 2.0 |
| KHN$_x$ | Group M | mg/L | Nitrogen half-saturation | 0.25 |
| | Group H1 | | | 0.25 |
| | Diatom | | | 0.45 |
| | Green | | | 0.45 |
| | Other | | | 0.45 |
| KHP$_x$ | Group M | mg/L | Phosphorus half-saturation | 0.10 |
| | Group H1 | | | 0.18 |
| | Diatom | | | 0.006 |
| | Green | | | 0.006 |
| | Other | | | 0.10 |
| TMX$_1$ | Group M | °C | Lower optimal temperature | 20.0 |
| | Group H1 | | | 10.0 |
| | Diatom | | | 2.0 |
| | Green | | | 2.0 |
| | Other | | | 20.0 |
| TMX$_2$ | Group M | °C | Upper optimal temperature | 35.0 |
| | Group H1 | | | 35.0 |
| | Diatom | | | 15.0 |
| | Green | | | 30.0 |
| | Other | | | 35.0 |
| WQRHOMN | Group M | kg/m$^3$ | Algae minimum density | 985 |
| | Group H1 | | | 920 |
| | Other | | | 970 |
| WQRHOMX | Group M | kg/m$^3$ | Algae maximum density | 1005 |
| | Group H1 | | | 1030 |
| | Other | | | 1065 |
| WQCOEF1 | Group M | kg/m$^3$/min | Rate constant of density increase | 0.030 |
| | Group H1 | | | 0.070 |
| | Other | | | 0.045 |
| WQCOEF2 | Group M | kg/m$^3$/min | Rate constant of density decrease | 0.0013 |
| | Group H1 | | | 0.001 |
| | Other | | | 0.001 |

| EFDC Parameter | | Unit | Definition | Nakdong River |
|---|---|---|---|---|
| WQCOEF3 | Group M | kg/ kg/m$^3$/min | Minimum rate of density increase | 0.013 |
| | Group H1 | | | 0.023 |
| | Other | | | 0.011 |
| WQR | Group M | m | Algae effective radius | 0.00008 |
| | Group H1 | | | 0.000005 |
| | Other | | | 0.00025 |
| CChl$_x$ | | mg C/µg Chl-a | Carbon-to-chlorophyll ratio for algae | 0.012 |
| CIa, CIb, Clc | | - | Weighting factor for solar radiationat 0, 1, and 2 d | 0.80, 0.15, 0.05 |
| BMR$_x$ | | /day | Basal metabolism rate for algae | 0.05–0.1 |
| PRR$_x$ | | /day | Predation rate on algae | 0.02 |
| CP$_{prm1}$ | | g C/g P | Constant for algae phosphorus-to-carbon ratio | 40 |
| CP$_{prm2}$ | | g C/g P | Constant for algae phosphorus-to-carbon ratio | 85 |
| CP$_{prm3}$ | | /mg/L | Constant for algae phosphorus-to-carbon ratio | 200 |
| ANC$_x$ | | g N/g C | Nitrogen-to-carbon ratio for algae | 0.18 |
| L_Factor1 | | W/m$^2$ | Convert light unit | 4.57 |
| F_PAR | | | Temperature and light average time | 0.44 |

**Table 5.** Model performance according to the parameter calibration.

| Group | Water Level (m) | Water Temperature (°C) | BOD (mg/L) | TN (mg/L) | TP (mg/L) | Chl-a (mg/m$^3$) |
|---|---|---|---|---|---|---|
| MAE * | 0.15 | 0.58 | 0.42 | 0.55 | 0.04 | 10.80 |
| RMSE ** | 0.25 | 0.71 | 0.55 | 0.63 | 0.05 | 13.47 |

* Mean Absolute Error. ** Root Mean Absolute Error.

## 3. Application and Result

### 3.1. Prediction of Harmful Cyanobacteria

Figure 7 compares the simulation results with the observed data collected 500 m upstream from the Changnyeong-Haman Weir. It is considered that the water quality items (e.g., nutrients and organic matters) that are highly related to the water flow characteristics (water level and water temperature), as well as the CyanoHABs and the behavior patterns of the weir section determining the CyanoHABs and its pattern, were reasonably reproduced (Table 5). The harmful cyanobacteria were simulated with the calibrated EFDC-NIER model from May to October, when CyanoHABs occurred, by converting the carbon content simulated for Group M (*Microcystis* spp.) and Group H1 (*Anabaena(=Dolichospermum)* spp. and *Aphanizomenon* spp.) using the carbon content per unit cell (pg C/cell) in Table 2. Figure 8 compares the observed and simulated harmful cyanobacteria data. The simulation predicted that harmful cyanobacteria occurred from May to October in the Changnyeong-Haman Weir. In particular, the number of harmful cyanobacteria cells was predicted to be approximately >100,000 cells/mL.

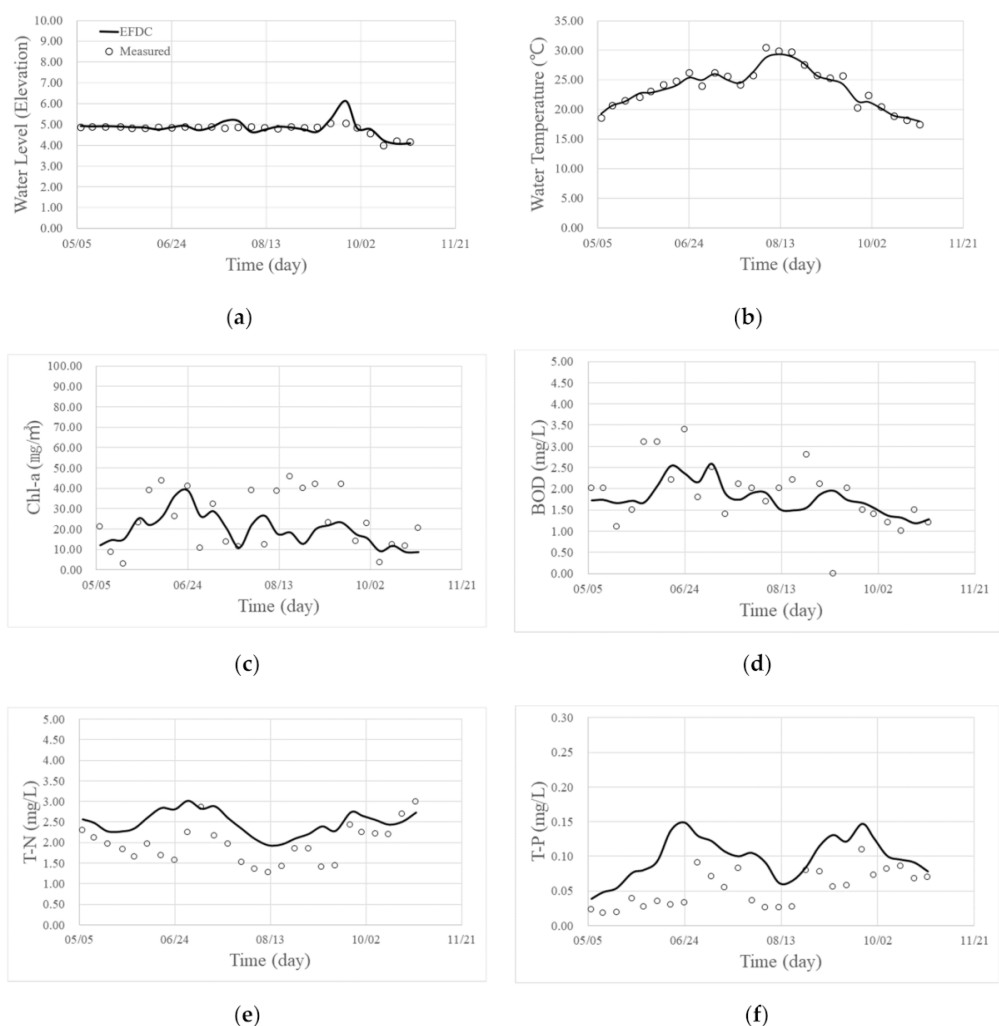

**Figure 7.** Comparison of simulated water quality results with those measured at a point 500 m upstream from the Changnyeong-Haman Weir: (**a**) Water level; (**b**) Water temperature; (**c**) Chlorophyll-a; (**d**) BOD; (**e**) T-N; (**f**) T-P.

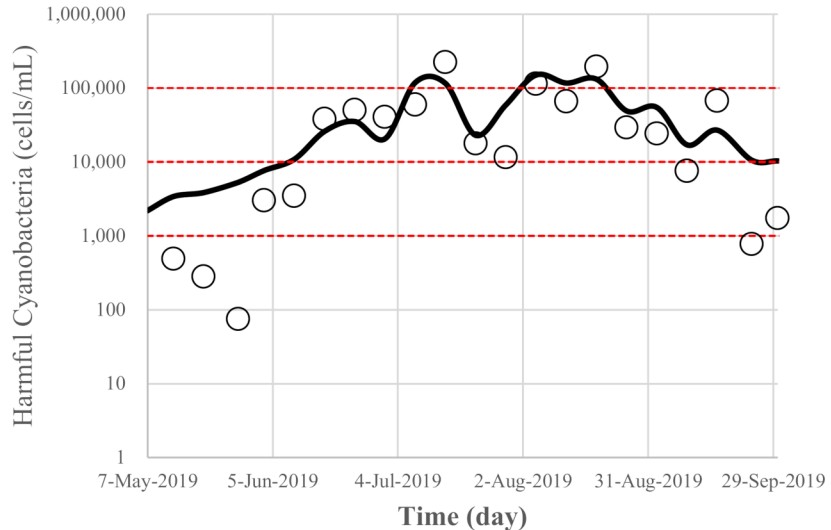

**Figure 8.** Simulated harmful cyanobacteria values compared with harmful cyanobacteria measured at a point 500 m upstream from the Changnyeong-Haman Weir.

*3.2. Applicability of the Model for Short-Term CyanoHABs Predictions*

The Harmful Algae Alert System of South Korea defines four alarm-triggering levels for harmful cyanobacteria based on their amount as follows: normal: <1000 cells/mL; concern: 1000–10,000 cells/mL; alert: 10,000–1,000,000 cells/mL; and bloom: >1,000,000 cells/mL [1]. However, the alert stage should be refined and subdivided because the range 10,000–1,000,000 cells/mL is too wide.

The World Health Organization and the Australian National Health and Medical Research Council have adopted water quality threshold values such as the "safe water quality for recreational activities" (harmful cyanobacteria < 20,000 cells/mL), "water quality that can cause adverse effects when humans or animals come in contact with the water" (20,000 < harmful cyanobacteria < 100,000 cells/mL), and "unsafe water quality when ingested by humans" (harmful cyanobacteria > 100,000 cells/mL) [25].

The South Korean Ministry of Environment operates a water quality monitoring network and a Harmful Algae Alert System based on observation points located 500 m upstream of eight multifunctional weirs. In the case of the Changnyeong-Haman Weir, which is a water source section, model predictions need to be provided that can determine if the water quality is unsafe for human ingestion based on harmful cyanobacteria counts (level 3: 10,000–100,000 cells/mL and level 4: 100,000–1,000,000 cells/mL). Table 6 lists the carbon occupancy proportions of harmful cyanobacteria measured at seven-day intervals from 2014 to 2019 at the water quality monitoring network point of the Changnyeong-Haman Weir. Over the five years, the harmful cyanobacteria amounts were less than 100,000 cells/mL for 25.2% of the time; for harmful cyanobacteria > 100,000 cells/mL, the occurrence proportion was 6.2%. Most of the harmful cyanobacteria observations (93.8%) were below 100,000 cells/mL. Given that the Harmful Algae Alert System in the existing section operates in the range 10,000–1,000,000 cells/mL, alert levels can often be triggered in the range 10,000–100,000 cells/mL. Once an alert is triggered, appropriate action should be taken. If two distinct alert levels (3 and 4) are used, step-by-step actions should be taken only if the concentration is 100,000 cells/mL or higher. In other words, a waste of administrative resources can be prevented by operating the Harmful Algae Alert System proactively and efficiently, that is, by using detailed CyanoHABs predictions and acting according to five distinct alert levels.

**Table 6.** Harmful cyanobacteria measured at the Changnyeong-Haman Weir.

| Level | Harmful Cyanobacteria Range (cells/mL) | Occupancy Proportion (%) | Cumulative | Harmful Cyanobacteria Occupancy Number |
|-------|----------------------------------------|--------------------------|------------|----------------------------------------|
| 1 | <1000 | 51.9 | 51.9 | 177 |
| 2 | 1000–10,000 | 16.7 | 68.6 | 57 |
| 3 | 10,000–100,000 | 25.2 | 93.8 | 86 |
| 4 | 100,000–1,000,000 | 6.2 | 100.0 | 21 |
| 5 | >1,000,000 | 0.0 | 100.0 | 0 |

In order to evaluate the applicability of the five distinct alert levels presented in this paper, the number of harmful cyanobacteria cells was predicted and the results were analyzed from May to October 2019. In South Korea, water quality and algae are measured every week. Therefore, the modified Harmful Algae Alert System proposed in this study was evaluated with a total of 26 data. The number of alert level (10,000–1,000,000 cells/mL) in existing Harmful Algae Alert System was 17, and 5 cases were analyzed for more than 100,000 cells/mL (Table 7). By comparing the results of the predictive modeling with the values reported in Table 8, the prediction was successful in 62% of cases. When evaluating the predictive power for the newly defined alert levels (3 and 4) by dividing them on the basis of 100,000 cells/mL, the prediction accuracy was analyzed as 65% (Table 7). The prediction accuracy is slightly reduced than before. However, considering that the

weather forecast accuracy of the Korean Meteorological Administration, which has the most significant effect on the CyanoHABs prediction, is approximately 70%. So, a success rate of 62% is considered the maximum possible goal. Consequently, a further increase in the success rate for the harmful cyanobacteria can be achieved by considering the modeling results for the entire year using the following equation:

$$\text{Success rate (\%)} = (A + B + C + D + E)/N \times 100 \tag{5}$$

where A, B, C, D, and E are the number when the observed and predicted data meet in each section; N = total number, as shown in Table 8.

**Table 7.** The evaluation result of predicted success rate.

| Date | Harmful Cyanobacteria (cells/mL) | | Harmful Algae Alert System | Modified Harmful Algae Alert System |
|---|---|---|---|---|
| | Observed Data | Predicted Data | | |
| 07.05.2019 | 0 | 2211 | Failure | Failure |
| 13.05.2019 | 490 | 3404 | Failure | Failure |
| 20.05.2019 | 282 | 3857 | Failure | Failure |
| 28.05.2019 | 75 | 5298 | Failure | Failure |
| 06.06.2019 | 3012 | 7661 | Success | Success |
| 10.06.2019 | 3461 | 10,877 | Failure | Failure |
| 17.06.2019 | 37,868 | 25,914 | Success | Success |
| 24.06.2019 | 50,432 | 35,339 | Success | Success |
| 01.07.2019 | 40,469 | 20,541 | Success | Success |
| 08.07.2019 | 59,526 | 118,145 | Failure | Success |
| 15.07.2019 | 223,562 | 115,711 | Success | Success |
| 22.07.2019 | 17,804 | 23,021 | Success | Success |
| 29.07.2019 | 11,540 | 58,737 | Success | Success |
| 05.08.2019 | 113,642 | 150,278 | Success | Success |
| 12.08.2019 | 66,145 | 117,259 | Failure | Success |
| 19.08.2019 | 194,924 | 131,850 | Success | Success |
| 26.08.2019 | 29,216 | 48,974 | Success | Success |
| 02.09.2019 | 24,274 | 54,836 | Success | Success |
| 09.09.2019 | 7573 | 17,010 | Failure | Failure |
| 16.09.2019 | 67,490 | 26,810 | Success | Success |
| 24.09.2019 | 777 | 10,542 | Failure | Failure |
| 30.09.2019 | 1738 | 10,305 | Failure | Failure |
| 07.10.2019 | 1738 | 3127 | Success | Success |
| 14.10.2019 | 1738 | 1429 | Success | Success |
| 21.10.2019 | 1738 | 1006 | Success | Success |
| 29.10.2019 | 1738 | 946 | Success | Success |

**Table 8.** Method for calculating the success rate of the harmful cyanobacteria prediction.

| Group | | Predicted Data | | | | |
|---|---|---|---|---|---|---|
| | | <1000 | 1000–10,000 | 10,000–100,000 | 100,000–1,000,000 | >1,000,000 |
| Observed Data | <1000 | A | - | - | - | - |
| | 1000–10,000 | - | B | - | - | - |
| | 10,000–100,000 | - | - | C | - | - |
| | 100,000–1,000,000 | - | - | - | D | - |
| | >1,000,000 | - | - | - | - | E |

## 4. Discussion and Conclusions

A method for predicting the amount of harmful cyanobacteria using the EFDC-NIER model was proposed. In addition, policy utilization measures, such as the operation of the Harmful Algae Alert System, were examined by predicting the harmful cyanobacteria

using the proposed method and the developed numerical tool. The major findings of this study are as follows:

1.  Harmful cyanobacteria occur in large amounts from June to August in Changnyeong-Haman Weir. According to National Institute on Environmental Research, the dominant algae observed from June to August in 2019 was *Microcystis* spp. Therefore, CyanoHABs are mainly caused by *Microcystis* spp. in South Korea. This phenomenon is demonstrated by the simulation results of harmful cyanobacteria in this study. The simulation focused on harmful cyanobacteria because it is the main cause of algal blooms. However, if the cell numbers of other relevant algal groups, such as diatoms and green algae, need to be predicted in the future, these algal groups can also be added. In addition, the carbon content (pgC/cell) simulated for each group and carbon content per unit cell in Table 2 can be converted to cell numbers. Therefore, detailed algal simulations for multiple species will be possible using the modeling method proposed in this study.

2.  The developed numerical tool was applied to the Hapcheon-Changnyeong Weir–Changnyeong-Haman Weir section of the Nakdong River, which experiences severe growth of HABs. The modified Harmful Algae Alert System subdivided the alert level (10,000–1,000,000 cells/mL) in existing Harmful Algae Alert System based on 100,000 cells/mL, which is a unsafe water quality when ingested by humans [25]. The total success rate for the prediction of harmful cyanobacteria was 62% (65% at level 3 and 4). The predictive power of the modified Harmful Algae Alert System presented in this study is slightly reduce, so it can be judged that its applicability is inferior. However, it is not true. For example, if the predicted number of harmful cyanobacteria cells was 900,000 cells/mL and the measured number was 20,000 cells/mL, the prediction is evaluated as successful in existing Harmful Algae Alert System. This is an overestimation of the predictive power due to the wide range of alert level in existing Harmful Algae Alert System. Because of these cases, predictive power should not be evaluated solely by predictive success rate. For this reason, it can be said that administrative power and budget were used inefficiently by over-reaction, even though it was not dangerous to humans. In other words, responding to the algal blooms problem by subdividing the sections according to severity like the modified Harmful Algae Alert System can prevent unnecessary administrative power and budget consumption, and manage algal blooms more efficiently. In terms of policy, more advanced algal blooms management will be possible if administrative power and budget wasted due to over-reaction are utilized elsewhere such as training experts, recruiting more researchers, increasing the algae measurement budget, and upgrading measurement and prediction equipment, etc. In this study, there is a limitation in that the sample size is small. Recently, algae are constantly being measured. In addition, the frequency of algae measurement has increased compared to before due to the advanced technique of estimating the concentration of harmful cyanobacteria using phycocyanin and the advancement of remote sensing such as hyperspectral image. For this reason, many high-quality data are being obtained. If high-quality data will be obtained more and more algal groups are considered in the future, the cell number simulation for multiple algal species as well as for harmful cyanobacteria will be possible, and it is foreseen that the success rate will also be improved.

Methods to ensure the model performs short-term CyanoHABs predictions were suggested. To establish the boundary conditions with respect to the inflow from tributaries into the studied river section, chlorophyll-a concentrations observed in the main stream were converted to monthly carbon proportions for each HAB prediction group. However, in order to apply scientifically-based boundary conditions, it is necessary to measure the cell number of each algal species at the inflows from the tributaries and convert them to a particular carbon content for each HAB prediction group. Furthermore, the initial conditions applied to the model domain are also important. If the initial harmful cyanobacteria concentrations are evaluated from the phycocyanin concentration observed

using hyper-spectroscopy (a remote sensing technique) rather than by linearly interpolating point-to-point data, the uncertainties can be reduced.

**Author Contributions:** Conceptualization, J.M.A.; data curation, J.M.A., J.K., L.J.P., J.J. (Jihye Jeon), J.J. (Jaehun Jong) and J.-H.M.; formal analysis, J.M.A.; funding acquisition, T.K.; investigation, J.M.A., J.K., L.J.P., J.J. (Jihye Jeon) and J.J. (Jaehun Jong); methodology, J.M.A. and J.-H.M.; project administration, J.M.A. and J.K.; resources, J.J. (Jaehun Jong); software, J.M.A.; supervision, J.K.; validation, J.M.A.; visualization, L.J.P.; writing—original draft, J.M.A.; writing—review and editing, J.K. and T.K. All authors have read and agreed to the published version of the manuscript.

**Funding:** This research was funded by the National Institute of Environmental Research (NIER), which is funded by the Ministry of Environment (MOE) of the Republic of Korea, grant number NIER-2020-01-01-012.

**Institutional Review Board Statement:** Not applicable.

**Informed Consent Statement:** Not applicable.

**Data Availability Statement:** The data presented in this study are available on request from the corresponding author.

**Acknowledgments:** This research was funded by the National Institute of Environmental Research of South Korea.

**Conflicts of Interest:** The authors declare no conflict of interest.

## Appendix A

National Institute of Environmental Research of South Korea monitors chlorophyll-a $(mg/m^3)$ and cyanobacteria (cells/mL) concentrations every seven days. Chlorophyll-a $(mg/m^3)$ is monitored by dividing the surface layer and water body mixing average, and cyanobacteria (cells/mL) is monitored at the surface layer. The detailed monitoring method used in this study is shown in Appendix A Figure A1. The sample of the surface layer was obtained by mixing the samples at three points on the left, center, and right located at 0.5 m water depth. The three points selected were the deepest point (center point), and two points separated by 1/4 of the total river width (along the cross section) to the left and right of the center point (left and right points). To obtain a representative sample that reflects the mixing average of the water body, samples from the same three points were taken. Additionally, samples were taken at 1/3 and 2/3 of the water depth from the surface. Then, the sample representative of the mixing average of the entire water body was obtained by mixing all collected samples.

Appendix A Figure A2 shows the results of chlorophyll-a $(mg/m^3)$ and cyanobacteria (cells/mL) observed from 6 January 2014 to 19 October 2020. In South Korea, water quality management was performed based on the sum of the number of cells monitored for the genera *Anabaena* spp., *Aphanizomenon* spp., *Microcystis* spp., and *Oscillatoria* spp. which release toxic compounds. Drinking water is contaminated due to the mass growth of *Microcystis* spp. from May to October in the Changnyeong-Haman Weir. Previously, the evaluation of CyanoHABs was performed using chlorophyll-a. However, chlorophyll-a is observed even when harmful cyanobacteria are not present, which is a significant issue for using it as an indicator. Appendix A Table A1 shows the monthly observations of harmful cyanobacteria and chlorophyll-a of the mixing average of the water body. Appendix A Figure A3 shows the correlation between harmful cyanobacteria and chlorophyll-a of the mixing average of the water body observed from 6 January 2014 to 19 October 2020. The correlation analysis shows that chlorophyll-a is not suitable for use as a representative index of CyanoHABs. Therefore, we attempted to develop a technique to manage CyanoHABs through the prediction of harmful cyanobacteria such as *Microcystis* spp.

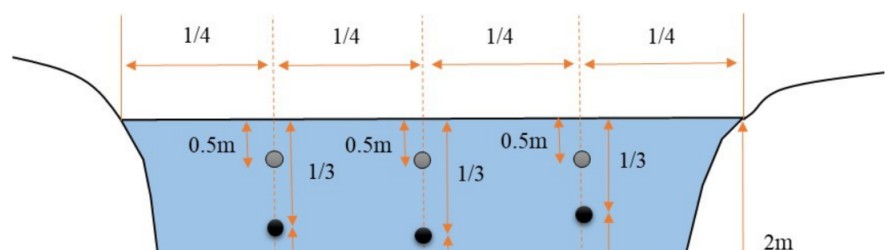

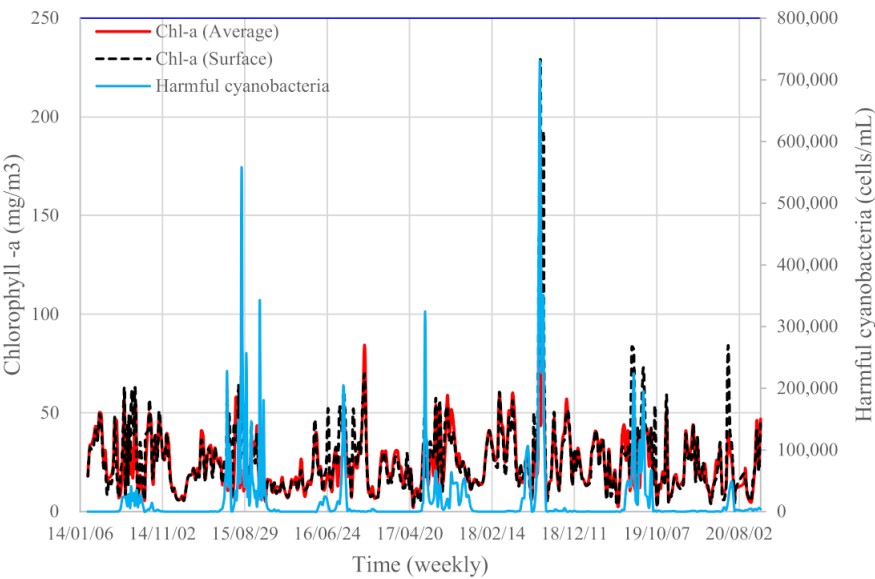

**Figure A2.** Comparison of harmful cyanobacteria and chlorophyll-a observed in the Changnyeong-Haman Weir.

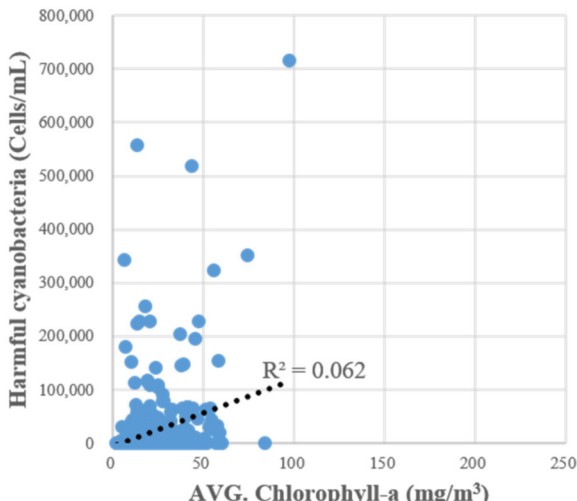

**Figure A3.** Correlation between average chlorophyll-a and harmful cyanobacteria.

**Table A1.** Monthly observed values of harmful cyanobacteria and chlorophyll-a.

| Month | Harmful Cyanobacteria (Cells/mL) | Chlorophyll-a of Water Body Mixing Average (mg/m$^3$) |
|:---:|:---:|:---:|
| 1 | 95 | 19.1 |
| 2 | 167 | 25.6 |
| 3 | 161 | 29.7 |
| 4 | 190 | 22.2 |
| 5 | 1181 | 22.3 |
| 6 | 44,443 | 22.8 |
| 7 | 30,647 | 25.1 |
| 8 | 118,064 | 28.4 |
| 9 | 26,984 | 27.0 |
| 10 | 21,697 | 27.4 |
| 11 | 15,609 | 24.5 |
| 12 | 882 | 14.1 |

**Appendix B**

Appendix B Table A2 shows an example of the calculation of the amount of carbon for each algal group and converting the calculated carbon amount to chlorophyll-a. β is the carbon/chlorophyll-a concentration ratio and its value is 0.12. For example, 0.2836 divided by 0.12 is 2.3633 for Codon P on 7 January 2019. The total chlorophyll-a value of all nine Codons from M to C is 14.94 (mg/m$^3$). Appendix B Figure A4 compares the observed and calculated chlorophyll-a values.

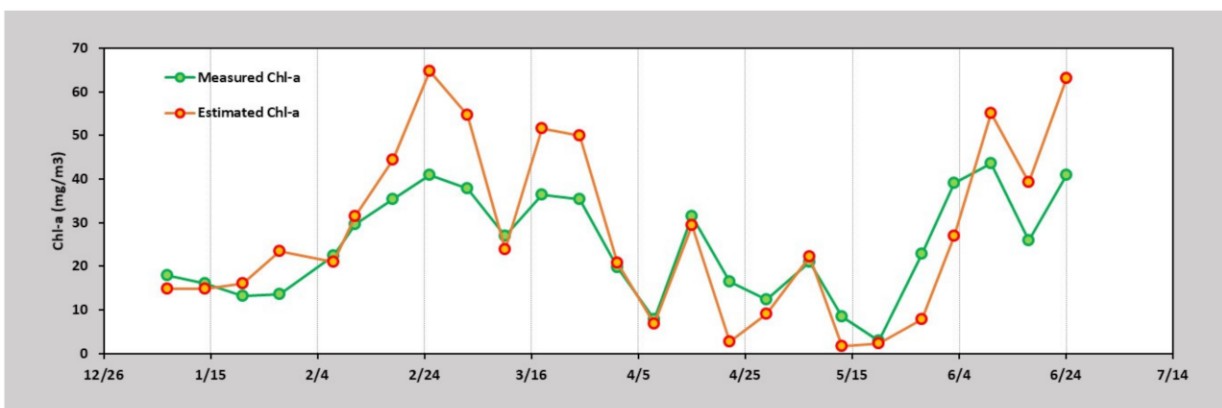

**Figure A4.** Comparison of measured and converted chlorophyll-a values.

**Table A2.** Examples of calculating the carbon amount for each algal group and converting it to chlorophyll-a.

| | | 07.01.2019 | 14.01.2019 | 21.01.2019 | 28.01.2019 | 07.02.2019 | 11.02.2019 | 18.02.2019 |
|---|---|---|---|---|---|---|---|---|
| Codon M | | 0.0000 | 0.0000 | 0.0000 | 0.0000 | 0.0000 | 0.0000 | 0.0000 |
| Codon H1 | | 0.0002 | 0.0000 | 0.0007 | 0.0039 | 0.0000 | 0.0010 | 0.0006 |
| Codon P | | 0.2836 | 0.2376 | 0.2276 | 0.2096 | 0.4900 | 0.6762 | 1.1559 |
| Codon D | | 1.0094 | 1.2031 | 1.0291 | 1.7233 | 1.4203 | 2.7011 | 3.8842 |
| Codon G | | 0.0000 | 0.0000 | 0.0022 | 0.0003 | 0.0000 | 0.0000 | 0.0005 |
| Codon X2 | | 0.1793 | 0.1886 | 0.3790 | 0.6776 | 0.4623 | 0.3472 | 0.2160 |
| Codon J | | 0.0005 | 0.0000 | 0.0000 | 0.0002 | 0.0000 | 0.0000 | 0.0009 |
| Codon LO | | 0.0000 | 0.0000 | 0.0000 | 0.0000 | 0.0000 | 0.0000 | 0.0224 |
| Codon C | | 0.3194 | 0.1573 | 0.2986 | 0.2077 | 0.1474 | 0.0454 | 0.0564 |
| ETC | | 0.0000 | 0.0000 | 0.0000 | 0.0000 | 0.0000 | 0.0000 | 0.0000 |
| SUM | | 1.7925 | 1.7866 | 1.9371 | 2.8226 | 2.5199 | 3.7708 | 5.3370 |
| Codon M | 0.12 | 0.0000 | 0.0000 | 0.0000 | 0.0000 | 0.0000 | 0.0000 | 0.0000 |
| Codon H1 | 0.12 | 0.0017 | 0.0000 | 0.0060 | 0.0322 | 0.0000 | 0.0082 | 0.0051 |
| Codon P | 0.12 | 2.3633 | 1.9801 | 1.8963 | 1.7467 | 4.0832 | 5.6348 | 9.6327 |
| Codon D | 0.12 | 8.4115 | 10.0257 | 8.5754 | 14.3608 | 11.8356 | 22.5089 | 32.3683 |
| Codon G | 0.12 | 0.0000 | 0.0000 | 0.0181 | 0.0023 | 0.0000 | 0.0000 | 0.0045 |
| Codon X2 | 0.12 | 1.4942 | 1.5718 | 3.1581 | 5.6468 | 3.8521 | 2.8930 | 1.7998 |
| Codon J | 0.12 | 0.0046 | 0.0000 | 0.0000 | 0.0018 | 0.0000 | 0.0000 | 0.0076 |
| Codon LO | 0.12 | 0.0000 | 0.0000 | 0.0000 | 0.0000 | 0.0000 | 0.0000 | 0.1870 |
| Codon C | 0.12 | 2.6620 | 1.3108 | 2.4887 | 1.7309 | 1.2282 | 0.3783 | 0.4699 |
| Total algae | | 14.94 | 14.89 | 16.14 | 23.52 | 21.00 | 31.42 | 44.48 |

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
