# Peer review of "Predicting Cyanobacterial Harmful Algal Blooms (CyanoHABs) in a Regulated River Using a Revised EFDC Model"

_water, doi:10.3390/w13040439_

Round 1

Reviewer 1 Report

My earlier comments have been adequately addressed.

Author Response

Thank you for your review. Thanks to your review, I and all authors were able to write the paper with high quality.

Reviewer 2 Report

There are still errors in the Table 4.

Please correct it as follows:

Algae minimum density:

WQRHOMN

Group M                0.985

Group H1              0.920

Other                    0.970

WQRHOMX           

Group M                1.005

Group H1              1.030

Other                    1.065

Reviewer 3 Report

Line 15: While I do not like references in abstracts, this sentence screams for one.

Line 16: Omit comma

Line 30: Omit comma, edit manuscript throughout for this type of punctuation error. I won't address the misuse of commas further.

Line 61: Should be ; thereby,

The Discussion-Conclusions Section is very weak and should be expanded. As it is, the manuscript is simply a review and application of a modeled process the emphasizes the results but not what the results mean to advancing the science and management of algal production. Without an expanded Discussion-Conclusions section, the manuscript is not significant enough for publication. This can be easily remedied by a well-referenced discussion section and a separate conclusions section (targeting what is next).

In general the paper is well-written and informative.

Author Response

This manuscript is a resubmission of an earlier submission. The following is a list of the peer review reports and author responses from that submission.

Round 1

Reviewer 1 Report

Comment:
The authors describe the development of a prediction model for cyanobacteria in a South Korean river system by calculating the approximate contribution of cyanobacteria to chlorophyll-a concentrations. This model is then tested to assess performance in predicting which management band for cyanobacteria the data point falls in. The study has merit, however, some changes are likely to be needed for the manuscript to appeal to a wide audience, these are detailed below. I have some major comments (below) that I think need to be addressed for the manuscript to be suitable for publication in Water. Additional comments and suggestions are included on the attached PDF.

I am particularly concerned by the lack of reference to the wider literature in large sections of the manuscript and would like to see this addressed, particularly in the introduction and discussion.

Comment 1:

As it currently stands, the manuscript is very focussed on the study area and river system in South Korea. I suggest the authors broaden the focus in the introduction and in the discussion areas of the manuscript to situate the manuscript within the wider international literature, which should improve its appeal to a broad audience. In particular, the authors should identify and highlight techniques or aspects of the models developed that could be helpful for water managers in other countries and river systems.

Comment 2:

The combined results and discussion section can be hard to follow and currently there is limited discussion of the results and their context within the wider literature within the field. It would be beneficial for the results and discussion sections to be separated, or at least each result and the corresponding discussion delineated more clearly in this section by breaking each result/discussion into separate paragraphs. Doing so would also help with comment 1 by enabling more discussion of the wider context of the study. Currently the layout makes it difficult to follow as several results and discussion points are interleaved within each paragraph. The results should be described in more detail, rather than just referring to the figures.

Comment 3:

A general overview schematic/diagram in the methods section explaining the workflow may be helpful for the reader for following the steps undertaken in developing and testing the models. In particular, it would be very helpful to show the dependencies for the different data inputs and at what point they interact with other variables in the model input.

Comment 4:

In general, figure and table captions should contain sufficient detail to stand alone or be interpretable without recourse to the methods or main text. Most of the figure and table captions require more information and particularly definition of abbreviations to assist the reader with interpreting these.

Comment 5:

Paragraph 2 (lines 44-59) is difficult to follow and it is not clear what point the authors are trying to lead the reader to. I recommend the authors consider decreasing some of the detail in this paragraph and re-structuring the information to highlight their key point that chlorophyll-a is used in models to predict cyanobacterial cell numbers.

Comment 6:

The term “cluster” for describing the different algal groups is unusual and is used somewhat interchangeably with algal group throughout this paragraph. Algal group (i.e. diatoms, green algae and cyanobacteria) is a term that is likely more intuitive for readers than cluster. I encourage the authors to consider choosing just one of these terms to facilitate ease of understanding for the reader.

Reviewer 2 Report

Comments to Authors:

This study describes development of a model relating water quality and phytoplankton composition to predict future harmful algal concentrations. The model was developed using data from a section of the Nakdong River during May to September 2019. The model accounted for 62% accuracy for prediction of HABs blooms and according to the authors would improve alert systems for HABs in South Korea.

The manuscript is generally well written and appropriately referenced. Some methodological issues need to be discussed and clarifications are required. The manuscript could be improved by providing wider context for interest by a wider group of readers. The data set (26 observations) is not sufficient for model development. As cast, there is a provincial feel to the narrative. Can the authors relate their work to other localities and/or provide examples that extend beyond a single growing season to test the model?

Specific Comments:

  1. 44 - This paragraph is not very clear. Given its importance in describing what the authors plan to execute, the clarity must be improved.

  1. 95 – Dividing the algal population into three clusters is an oversimplication of the phytoplankton community.

  1. 113-120 – A number of these variables are not commonly collected in water quality studies, especially the variables listed in lines 117-120.

L.251 – Planktonic Anabaena is now known as Dolichospermum.

  1. 303 – 26 data points is a very limited group of data.

  1. 378 – What is the criteria for potential toxigenicity. Be specific.

Reviewer 3 Report

Predicting of harmful blooms of cyanobacteria is very important issue especially when we have to cope with increasing eutrophication caused by inflow into rivers and waterbodies the agricultural fertilizers and not adequately treated wastewater. As toxins producing cyanobacteria could cause health, economic and environmental problems all over the world it is urgent need to develop tools ready to apply in water environmental management. That is why EFDC-NIER model proposed by the authors seems to be very useful as dveloped basing on many parameters and data. The relatively high success rate for the prediction of harmful cyanobacteria bloom is promising. I recommend the article for publication in Water after minor revision. A few comments are listed below.

Page 1, 20 – add “South Korea”

Page 1, 31  – not always “green”

Page 1, 36 – replace first “and” with “,”

Page 2. 62 – delete “in”

Page 2, 83 – “to predict” instead of “prediction”

Page 3, 104-106 – I suggest to insert meaning near the abbreviation

Page 3, Fig 1 – in the figure caption describe colors meaning

Page 3, 113-115 – when used for the first time in the paper all abbreviation should be explained

Page 4,140 – please clarify the way of calculation for carbon content per liter for each species

Page 5, 164 – number of equation (2) should be placed in similar pattern as in case of equation 1

Page 6, Fig. 3 – Copylight?

Page 7, 210 – Figure 3c?

Page 9, 38-39 – delete mistakenly added brackets

Table 4  – please carefully check all units (some values are improbable) and improve formatting

Table 7 – please clarify the method in the table caption

Round 2

Reviewer 2 Report

The revised manuscript addresses my comments.

Author Response

Thanks to your review, the paper has been revised well.

Reviewer 3 Report

Table 4  – please carefully check all units (some values are improbable) and improve formatting

  • There is no problem with the unit in the table. As reviewers recommended, we have improved the format of the table.

Rev: It is impossible to reach such values as 1065 kg/m3. The mass of 1 m3 of water equals 1000 kg. Did you mean 1.065 kg/m3 or 1065 g/m3?

kg/m3

Algae minimum density

985

920

970

kg/m3

Algae maximum density

1,005

1,030

1065

Author Response

Thank you for your review. We carefully checked all units and improved table 4.

The value as 1065 kg/m3 is typing error. 1,065 kg/m3 is correct.

So, we modified it (see Table 4).

The EFDC parameter WQRHOMX refers to the maximum density of algae.

Therefore, the unit as kg/m3 is correct.

For example, the density of water at 4 degrees celsius is 1,000 kg/m3.

Likewise, WQRHOMX (maximum density of algae) of other algae species is 1,065 kg/m3